# CD44, PDL1, and ATG7 Expression in Laryngeal Squamous Cell Carcinomas with Tissue Microarray (TMA) Technique: Evaluation of the Potential Prognostic and Predictive Roles

**DOI:** 10.3390/cancers15092461

**Published:** 2023-04-25

**Authors:** Lidia Puzzo, Maria Rita Bianco, Lucia Salvatorelli, Giordana Tinnirello, Federico Occhiuzzi, Daniele Latella, Eugenia Allegra

**Affiliations:** 1Department “G.F. Ingrassia”, Section of Anatomic Pathology, University of Catania, 95123 Catania, Italy; lipuzzo@unict.it (L.P.);; 2Department of Medical and Surgical Sciences, Otolaryngology, University of Catanzaro, 88100 Catanzaro, Italy; mrbianco@unicz.it; 3Department of Health Science, Otolaryngology, University of Catanzaro, 88100 Catanzaro, Italy

**Keywords:** biomarkers, laryngeal cancer, CD44, PDL1, ATG7, prognosis

## Abstract

**Simple Summary:**

We are going to highlight new prognostic and predictive factors, CD44, PDL1, and ATG7, in surgical samples from patients with laryngeal squamous cell carcinoma (LSCC). Immunohistochemical analysis with primary antibodies anti-CD44, PD-L1 and ATG7 was performed using the tissue microarray (TMA) technique. Considering follow-up data, the 5-year disease-free survival (DFS) was 85.71% and 36% for CD44-negative and -positive tumors. The 5-year DFS was 60% and 33.33% for negative and positive PDL1 tumors, respectively. At the follow-up, 5-year DFS was 58.06% and 37.50% for negative and positive ATG7 tumors.

**Abstract:**

We focus on the new prognostic and predictive factors CD44, PDL1, and ATG7 in our study of surgical samples of patients with laryngeal squamous cell carcinoma (LSCC) using tissue microarray (TMA). Thirty-nine previously untreated patients affected by laryngeal carcinoma who then underwent surgical treatment were considered in this retrospective study. All surgical specimens were sampled, embedded in paraffin blocks, and stained with hematoxylin and eosin. A representative sample of the tumor was chosen and transferred into a new block of paraffin, the recipient block, to perform immunohistochemical analysis with the primary antibodies anti-CD44, PD-L1, and ATG7. At follow-up, 5-year disease-free survival (DFS) for negative and positive tumors was determined as 85.71% and 36% for CD44, 60% and 33.33% for PDL1, and 58.06% and 37.50% for ATG7, respectively. Multivariate analysis revealed that CD44 expression is an independent predictive factor of low-grade tumors (*p* = 0.008), lymph node metastasis at the time of diagnosis, and AGT7 negativity. Thus, CD44 expression is a potential marker for more aggressive forms of laryngeal cancer.

## 1. Introduction

Laryngeal cancers represent one-third of head and neck cancers and 4.5% of all malignancies [1]; more than 90% of all malignant neoplasms in the larynx are squamous cell carcinomas (SCCs) with multiple histologic variants and variable behavior. Head and neck carcinomas include tumors of various sites in which treatment modalities widely differ, not only in relation to the site but also clinical staging. The treatment response varies from individual to individual in terms of the biological behavior of the primary tumor. Currently, the choice of treatment is dependent on the disease stage at the time of diagnosis and includes different surgical and therapeutic modalities, including transoral laser microsurgery [2], organ partial horizontal laryngectomy [3], and radiotherapy, with good oncological and functional outcomes. In recent years, the incidence of open preservation surgery has increased; with open partial laryngectomy representing a particularly valuable alternative to total laryngectomy after chemoradiotherapy failure and transoral laser surgery recurrence. The purpose of this surgery is to avoid highly mutilating total laryngectomy [4] and, with respect to oncological radicality, to maintain laryngeal function [5]. At the time of diagnosis, most LSCC patients present with an advanced stage of the disease for which total laryngectomy is required in treatment [6]. Regarding patient quality of life, total laryngectomy has important psychophysical and social consequences due to functional changes in vocalization resulting from the removal of the larynx, leading to an immediate loss of phonatory function [7] and permanent separation of the upper from the lower airways for which a tracheostomy is required to allow patients to breathe [8]. The evolution of LSCC cannot be clinically determined based on current staging criteria, which fail to allow differentiation between those patients likely to experience metastasis, recurrence, or a second tumor from those who are not. There are many ongoing studies attempting to identify genetic factors and oncogenes that could be of etiological, prognostic, and predictive interest for LSCC. Recently, new therapies based on the molecular characteristics of SCC are under development. Tumor markers are considered new therapeutic targets used to block specific pathways involved in cancerization and tumor aggressiveness or invasiveness and to restore oncosuppressor activities. Here, we highlight the discovery of new prognostic and predictive factors, CD44, PDL1, and ATG7, from our study of surgical samples of patients with laryngeal squamous cell carcinoma (SCC) using tissue microarray (TMA). CD44 is a transmembrane glycoprotein expressed in several cells of mesenchymal and neuroectodermal origin. CD44 functions as a major adhesion molecule and in the cellular internalization of hyaluronic acid in both normal and cancer cells. Programmed cell death 1 (PD1) and its ligand (PDL1) are immune checkpoints that are able to control the tumoral microenvironment and inhibit immune responses against tumor cells. PDL1 is physiologically expressed by a subset of macrophages and can be induced in lymphocytes, endothelial cells, and the inflammatory microenvironment. It can be overexpressed in tumor cells and inflammatory stromal cells of the tumor microenvironment; therefore, anti-PDL1 monoclonal target therapy could result in blocking this mechanism and restoring immune responses against tumor cells. PDL1 expression in tumor cells and peri- and intratumoral cytotoxic T lymphocytes could be considered a predictive factor for biological target therapy. Autophagy-related genes (ATGs) produce autophagic pathway regulatory proteins; their role is related to the production of phagocytic organelles in physiological and pathological conditions. Autophagic activity facilitates therapy resistance and the survival of dormant cells in metastatic sites, maintaining a dormant phenotype and radio-/chemoresistance. Among the many autophagic proteins, ATG7 is used as a marker due to its behavior and involvement in oncogenesis.

## 2. Materials and Methods

Thirty-nine previously untreated patients affected by laryngeal carcinoma who then underwent surgical treatment between January 2009 and December 2014 at the Otolaryngology Department of Health Sciences, University of Catanzaro, were considered in this retrospective study. Patients with metastatic disease or with synchronous tumors at the time of diagnosis were excluded. For each patient, clinical–anamnestic data, including of age, sex, primary tumor site, histopathology, TNM, and follow-up, were collected in a database. The stage was determined in accordance with the 8th edition of the TNM classification established by the American Joint Committee for Cancer. Data on smoking habits were also collected, defining nonsmokers as those who had never smoked or had quit, and smokers as those who had smoked regularly before or were still active smokers. Patients were asked how many cigarettes they smoked from which a pack/day figure was then calculated. We defined a light/moderate smoker as someone who smoked ≤20 cigarettes/day and a heavy smoker as someone who smoked >20 cigarettes/day. Patients who indicated that they drank alcohol regularly were defined as alcohol drinkers. We considered a nonalcohol drinker as someone who never drank alcohol or only drank on very rare occasions, consuming less than 200 cc per day overall, a light–moderate drinker as someone who drank ≤1000 cc/day, and a heavy drinker as someone who drank >1000 cc/day. Finally, we recorded the data related to any disease recurrence and locoregional or distant metastases detected during follow-up. All patients included in this study were followed up every three months. Due to its retrospective nature of evaluation, our study did not require any authorization by the Institutional Review Board. All surgical specimens were sampled, embedded in paraffin blocks, and stained with hematoxylin and eosin, and histologic diagnosis was made by two expertise pathologists (L.P., L.S.). For each block, a representative sample of the tumor was chosen and transferred into a new block of paraffin, termed the recipient block, according to the procedures required for tissue microarray (TMA) [9]. From the recipient block, one slide was stained with hematoxylin and eosin, and the other 3 µm sections were cut and placed onto pretreated slides to perform immunohistochemical analysis with primary antibodies anti-CD44 (Abcam; clone F10-44-2; dil. 1:200), PD-L1 (Dako; Clone 22C3; PHARM Dx), and ATG7 (Abcam; clone EP1759Y; dil. 1:500). The immunoreactive score (IRS) was used as a semiquantitative method to assess the biomarkers’ expression [10]; it is the sum of the ordinal scores for distribution and intensity of immunostaining. The immunoreactive score (IRS) of CD44 and ATG7 expression was calculated as the sum of the percentage of positive cells (range 0–4) and their immunostaining intensity (range 0–3). The IRS range was 0–12, where the samples ranging 0–5 were considered low IRS (L-IRS) and those in the range 6–12 considered high IRS (H-IRS). With regard to PDL1 expression, the combined positive score (CPS) was calculated as the number of PDL1-positive tumor cells, lymphocytes, and macrophages divided by the total number of tumor cells and multiplied by 100. CPS > 1 is considered positive and predictive of potential immunotherapeutic response [11].

## 3. Statistical Analysis

Statistical analyses were performed using MedCalc software 20.113 (Belgium) for chi-squared and Fisher’s exact tests. The collected data are described in terms of means, medians, and standard deviations. Pearson’s chi-square or Fisher’s exact tests were used to identify differences in demographic and clinicopathologic data between cohorts. Variables considered in the survival analysis include age, T and N stage, adjuvant therapy, tumor subsite, alcohol and smoking habits, and biomarker expression. Multivariate analysis was performed using multiple regression analysis to determine independent prognostic factors. Correlations between combined markers and clinical data were detected using the Mann–Whitney *U*-test. *p* < 0.05 was considered to indicate statistically significant differences. Disease-free survival (DFS) was calculated according to the Kaplan–Meier method.

## 4. Results

### 4.1. Patients’ Clinical Pathohistological Data

The median age of patients was 64.7 ± 10.4 SD years (range 46–83), and the mean follow-up time was 77.38 ± 48.2 SD. All patients were smokers: 22/39 (56.4%) were light–moderate smokers and 17/39 (43.6%) were heavy smokers. Moreover, 4 of the 39 (10.2%) patients were nondrinkers and 35/39 (89.8%) were drinkers of which 14 (40%) were light drinkers and 21 (60%) were heavy drinkers. With regard to the site, 5/39 (12.8%) tumors were transglottic, 10/39 (25.6%) were supraglottic, and 24/39 (61.6%) were glottic. The histological grade was G1 in 1/39 (2.6%) patients, G2 in 23/39 (58.9%) patients, and G3 in 15/39 (38.5%) patients. With regard to staging, 28/39 (71.7%) patients were T1–T2 stage, 11/39 (28.3%) patients were T3–T4 stage. The lymph node status was N0 in 26/39 (66.6%) patients and N+ in 13/39 (33.3%) patients. A total of 27/39 (69.3%) patients underwent neck dissection, and 20/39 (52%) patients were subjected to radiotherapy after surgery (Table 1). When considering follow-up data, 22 patients (56.4%) were disease-free during follow-up, while distant metastasis to the lung, liver, esophagus, bone, and distant lymph nodal sites occurred in 16 patients (41%); 1 patient (2.6%) died from stroke. The patients’ clinical data were stratified according to CD44, PDL1, and ATG7 expression and were evaluated using the Mann–Whitney test (Table 2).

### 4.2. CD44 Expression

In 36 of the 39 samples has been possible to obtain evaluable immunohistochemistry staining. With regard to CD44 expression, 25/36 (69.4%) cases showed CD44 expression and 14/36 69.4%) were negative. Of the 25 CD44-positive cases, 14 (56%) were considered H-IRS (Figure 1 and Figure 2), and, in 7 of these cases, there was locoregional lymph node metastases at the time of diagnosis; 11 (44%) were considered L-IRS, and, in 5 of these cases, there was locoregional lymph node metastases at the time of diagnosis. With regard to negative cases, there was locoregional lymph node metastases at the time of diagnosis in only 1/11 (9%) cases. A total of 10 out of the 14 (71.4%) H-IRS cases and 9/11 L-IRS cases (81.8%) were classified as G1-G2 grade following histological evaluation; only one G1 tumor was CD44-negative. During follow-up, the 5-year disease-free survival (DFS) was 85.71% and 36% for CD44-negative and -positive tumors (*p* = 0.008; Figure 3), and CD44 expression was detected in 9 of the 16 patients (56.2%) with distant metastasis of which 4/9 (44.4%) cases were considered H-IRS and 5/9 (55.5%) L-IRS. When stratifying the clinical data according to the CD44 expression, the results of Mann–Whitney statistical analysis confirmed a significant correlation between pN status (*p* = 0.04) and histologic grade (*p* = 0.0065) (Table 2).

### 4.3. PDL1 Expression

PDL1 expression was detected in 9/39 (23.0%) cases (Figure 4 and Figure 5) of which 7/9 (77.7%) cases were classified as T1–T2 stage, with no regional lymph nodal metastases at the time of diagnosis in 6 of these cases; 5/9 (55.5%) were G1–G2 grade and 4/9 (44.4%) were G3 grade. During follow-up, PDL1 expression was detected in 4/22 (18.2%) disease-free patients and in 5/16 (31.2%) patients with distant metastasis to the lung, liver, esophagus, bone, and distant lymph nodal sites. The 5-year DFS was 60% and 33.33% for negative and positive PDL1 tumors, respectively (*p* = 0.22; Figure 6). When stratifying the clinical data according to PDL1 expression, no significant correlations were noted (Table 2).

### 4.4. ATG7 Expression

ATG7 expression was evaluable in 36 samples. With regard to ATG7 expression, 8/36 cases (22.2%) were considered L-IRS and none as H-IRS (Figure 7), and 5/8 (62.5%) of these L-IRS patients presented with an advanced stage of disease at the time of diagnosis, where 3/8 (37.5%) were G2 grade and 5/8 (62.5%) were G3 grade. During follow-up, ATG7 expression was detected in 22.7% disease-free patients and in 18.7% patients with distant metastasis, and the 5-year DFS values were 58.06% and 37.50% for negative and positive ATG7 tumors, respectively (*p* = 0.483; Figure 8). No significant correlations were noted after stratifying the clinical data according to the ATG7 expression.

### 4.5. Correlation between the Expression of Biomarkers

We evaluated the correlations between the expressions of all markers. The most significant result was found for the relationship between CD44 expression and ATG7 negativity (*p* < 0.02). Multivariate analysis regarding clinical data and biomarker expression revealed that CD44 expression is an independent predictive factor of low-grade tumors (*p* = 0.008), lymph node metastasis at the time of diagnosis (*p* = 0.01), and AGT7 negativity (*p* = 0.049) (Table 3).

## 5. Discussion

The aim of this study is to evaluate the role of new prognostic and predictive markers in laryngeal carcinomas. CD44, PDL1, and ATG7 immunohistochemical expression and clinical data were compared and analyzed, focusing on CD44 expression. CD44 is a transmembrane glycoprotein involved in aggregation, proliferation, and migration in both normal and cancer cells and also in angiogenesis [12]. Furthermore, recent studies have highlighted CD44 as an important biomarker of cancer cell subpopulations and cancer stem cells (CSCs), exhibiting characteristics of self-renewal and tumor initiation, progression, invasion, metastasis, and recurrence, as well as chemo-/radiotherapy resistance. There are many discrepancies in the interpretation of CD44 expression in head and neck carcinomas, exhibiting different biological characteristics depending on their subsites. As previously reported for the oral cavity and oropharynx, low CD44 expression is correlated with increasing metastasizing behavior and poor prognosis [13,14], while in laryngeal carcinomas, high CD44 expression is correlated with increasing metastasizing behavior and radiotherapy resistance [15,16,17]. There are few data on the prognostic role of CD44 expression in laryngeal carcinomas. In a systematic review and meta-analysis of CD44 expression in head and neck cancers, Chen et al. [18] reported CD44 expression in 54.7% laryngeal carcinomas. In our previous paper [19] on the prognostic role of salivary CD44sol levels during follow-up of laryngeal carcinomas, high levels of salivary CD44sol were found in patients with laryngeal carcinoma compared with controls, mainly in the case of advanced-stage disease with lymph node metastasis at the time of diagnosis. Our results here reveal CD44 expression in 64.1% of laryngeal carcinomas, almost half of which exhibited lymph node metastases at the time of diagnosis, confirming its role in the selection of cancer stem cells, inducing disease progression and therapy resistance. Clusters of cancer stem cells could spread from the primary tumor site to distant sites, where new tumor niches could maintain a dormant state for months or years and develop resistance to therapy. These results suggest the potential prognostic role of CD44 expression as a marker of more aggressive laryngeal carcinoma independent of its grade and stage; high CD44 expression was observed in low-grade (G2) and advanced-stage carcinomas with lymph node metastasis at the time of diagnosis. In addition, CD44 expression can be detected in the majority of patients (56%) with distant metastasis during follow-up. With regard to PDL1 expression in laryngeal carcinomas, there are few studies in the literature, and their results are mostly conflicting. In a recent study, Wusiman et al. [20] evaluated PDL1 expression in head and neck squamous cell carcinomas, including all sites and considering the potential relation with clinicopathologic characteristics, mainly via the use of follow-up data. Other previous studies showed a correlation between PDL1 expression and both poor and good prognosis, and this discrepancy may be due to the different evaluation criteria used for PDL1 expression [21,22]. Nevertheless, the role of PDL1 expression has been linked to the possibility of tumor cells evading immunosurveillance by the immune system, mainly when there is PDL1 expression both in cancer cells and in inflammatory cells of the tumor microenvironment. Anti-PDL1 target therapy has already been approved for advanced-stage lung and breast carcinomas where PDL1 expression was evaluated using an international validated protocol. At present, several technical criticisms have been raised regarding the use of CPS as an indicator of PDL1 expression in head and neck carcinomas, which are related to the different clones of the PDL1 antibody used for lung carcinomas and the poorly defined cut-off level for positivity. Moreover, in 2019, the FDA approved use of a new anti-PDL1 target therapy, both in isolation or in combination with another conventional therapy, for advanced-stage head and neck squamous cell carcinomas [23,24]. In our study, PDL1 expression was only detected in a few cases (23%) irrespective of stage and follow-up but often associated with CD44 expression. It is interesting to note that advanced-stage carcinomas with lymph node metastasis at the time of diagnosis that underwent radio/chemotherapy before surgical treatment developed metastatic disease during follow-up anyway. This unfavorable prognostic behavior could be related to PDL1 expression, and anti-PDL1 target therapy could be beneficial in such cases. There are currently few studies in the literature regarding ATG7 expression [25], mainly in head and neck squamous cell carcinomas [26,27]. In our preliminary study, we found low representation of ATG7 expression in laryngeal carcinomas (22.2%). No correlation was found with clinical data, but a significant inverse correlation was found between CD44 expression and ATG7 negativity; in 84.6% of CD44-positive N+ tumors, no AGT7 expression was detected. These results can be explained by the contrasting activity of biomarkers; in fact, CD44 expression promotes cell proliferative activity in contrast to the “dormant cell” status promoted by AGT7. Furthermore, Pramatik et al. demonstrated that CD44-expressing tumors represent a great potential target for an effective therapy with Hyaluronic-Acid-Tagged Cubosomes [28]. However, further studies with a larger number of cases are needed to clarify this hypothesis. Our study presents some limitations due to the limited number of patients taken into consideration and the fact that it is retrospective in nature. However, the strength of this study lies in the homogeneity of the selected patients with all presenting laryngeal cancer and being subjected to surgical treatment.

## 6. Conclusions

CD44 expression is a potential marker for more aggressive forms of laryngeal cancer. CD44 expression, together with negative ATG7 expression, seems to characterize a subset of patients with higher risk of developing lymph nodal and distant metastasis. PDL1 could also be a biomarker for this cancer, although it is not as effective as CD44. Further studies are needed to confirm our preliminary results.

## Figures and Tables

**Figure 1 cancers-15-02461-f001:**
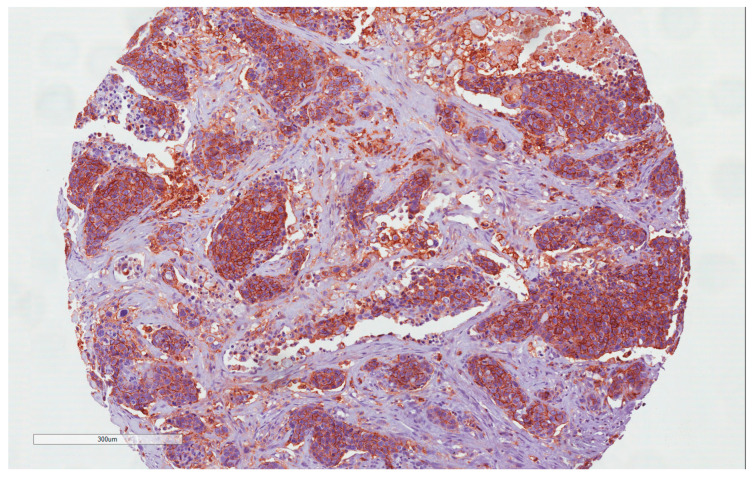
CD44 immunohistochemistry staining: H-IRS in tumor cells (10×).

**Figure 2 cancers-15-02461-f002:**
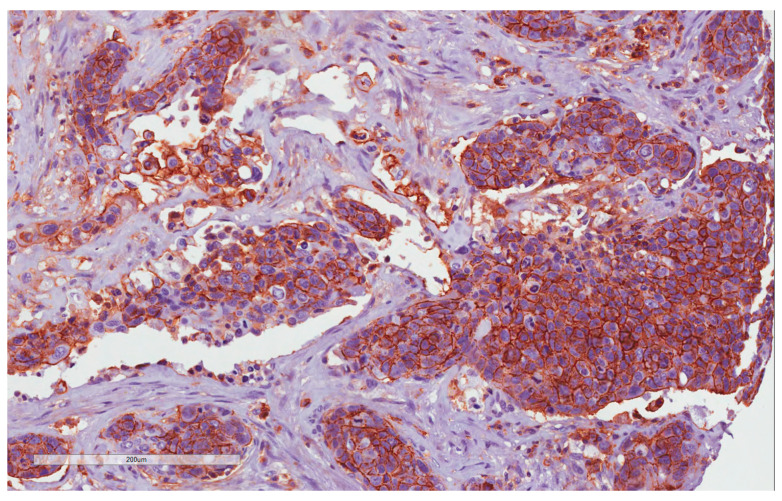
CD44 immunohistochemistry staining: H-IRS in tumor cells (20×).

**Figure 3 cancers-15-02461-f003:**
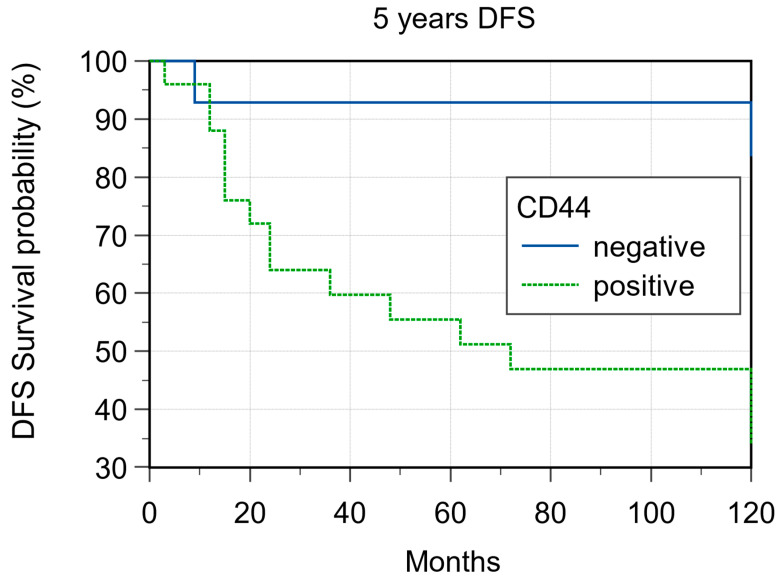
Five-year disease-free survival according to CD44 expression (*p* = 0.008).

**Figure 4 cancers-15-02461-f004:**
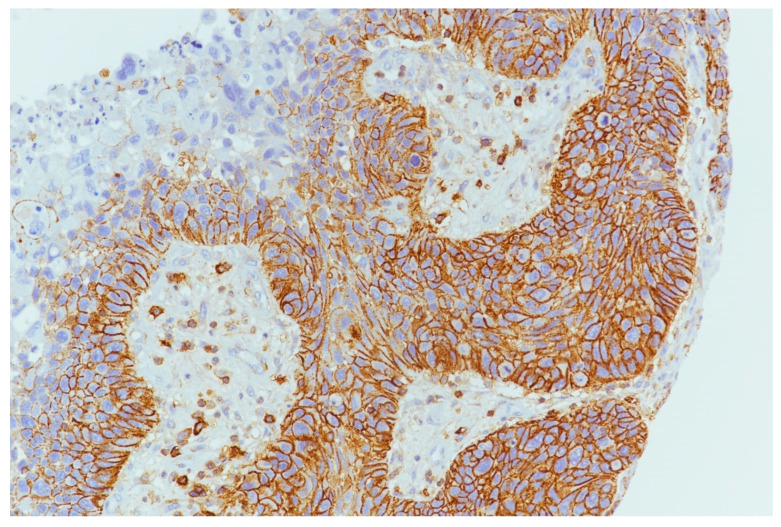
PDL1 expression in tumor cells and inflammatory cells in microenvironment (10×).

**Figure 5 cancers-15-02461-f005:**
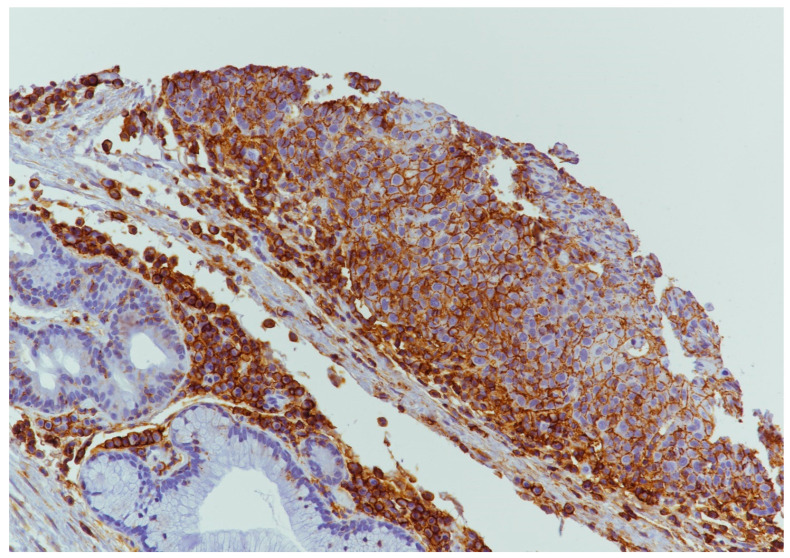
PDL1 expression in tumor cells and inflammatory cells in microenvironment (20×).

**Figure 6 cancers-15-02461-f006:**
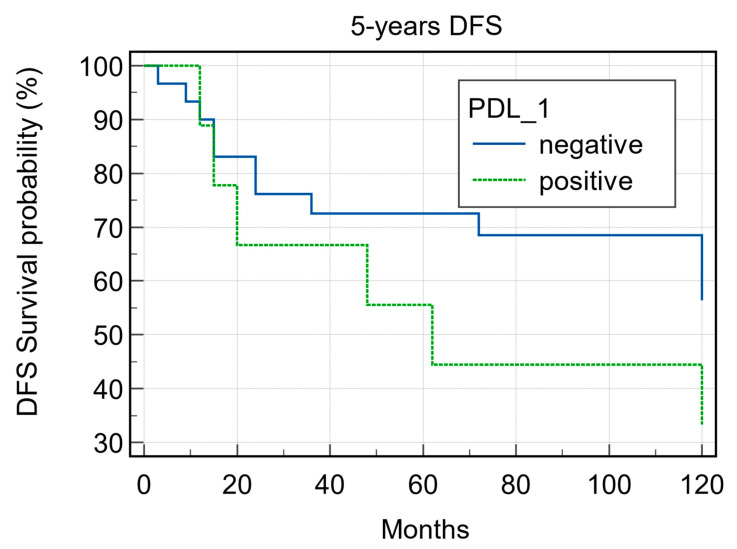
Five-year disease-free survival according to PD1 expression (*p* = 0.22).

**Figure 7 cancers-15-02461-f007:**
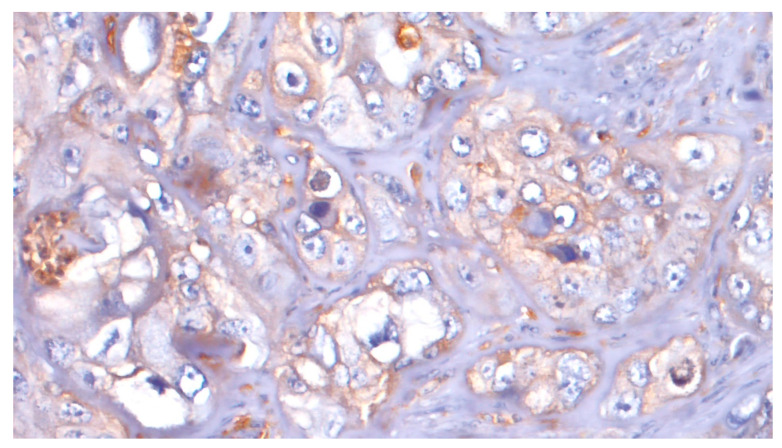
ATG7: L-IRS in tumor cells (20×).

**Figure 8 cancers-15-02461-f008:**
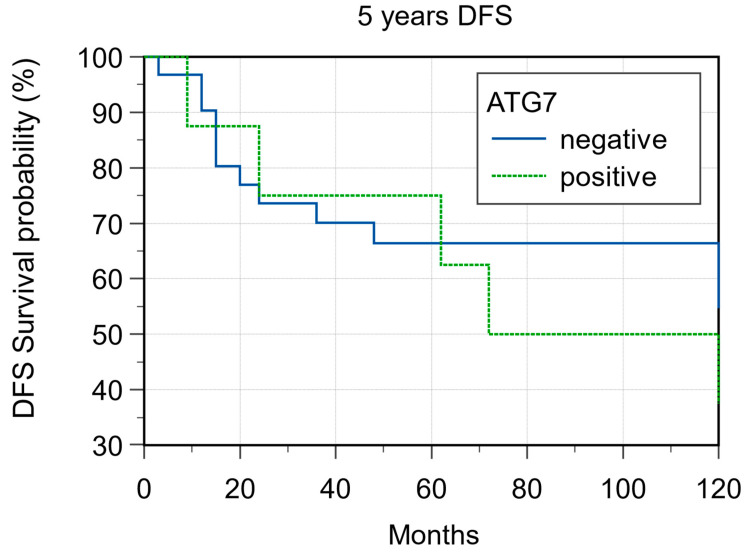
Five-year disease-free survival according ATG7 expression (*p* = 0.48).

**Table 1 cancers-15-02461-t001:** Patient’s clinical-anamnestic data.

Clinical Data	N. (%)
Age	
Mean	64.7 ± 10SD
range	46–83 yrs
Smoking habit	
≤20 cig/day	22
>20 cig/day	17
Alcohol habit	
≤500 cc/day	18
>500 cc/day	21
Primary Site	
Glottic	24
Trans-Supraglottic	15
Histologic grade	
G1–G2	24
G3	15
T classification	
T1–T2	28
T3–T4	11
N pathological status	
N0	26
N+	13
Treatment	
Surgery alone	19
Surgery plus RT	20

**Table 2 cancers-15-02461-t002:** Patients’ distribution according to biomarker expression.

Clinical Data	CD44 = 36 pz				PDL1 = 39 pz				AGT7 = 36 pz			
	Negative		Positive		Negative		Positive		Negative		Positive	
	N. (%)		N. (%)		N. (%)		N. (%)		N. (%)		N. (%)	
Age (years)												
≤50	3	(27.2%)	8	(72.8%)	12	75.0%	4	30.5%	11	68.5%	5	31.5%
>50	8	(32.0%)	17	(68.0%)	18	78.3%	5	21.7%	17	85.0%	3	15.0%
				*p* = 0.30				*p* = 0.31				*p* = 0.12
Smoking habit												
≤20 cig./day	7	33.3%	14	66.4%	18	81.8%	4	18.2%	15	75.0%	5	25.0%
>20 cig./day	4	26.6%	11	73.4%	12	70.6%	5	29.4%	13	81.3%	3	18.8%
				*p* = 0.34				*p* = 0.81				*p* = 0.78
Alcohol habit												
≤1000 cc/day	6	37.5%	10	62.5%	19	86.4%	3	13.6%	12	75.0%	4	25.0%
>1000 cc/day	5	25.0%	15	75.0%	12	66.7%	6	33.3%	16	80.0%	4	20.0%
				*p* = 0.42				*p* = 0.38				*p* = 0.72
Localization												
Glottic	8	36.3%	14	63.4%	18	75.0%	6	25.0%	17	77.3%	5	22.7%
Supra/transglottic	3	21.4%	11	78.6%	12	80.0%	3	20.0%	11	78.6%	3	21.4%
				*p* = 0.76				*p* = 0.72				*p* = 0.46
Histologic grade												
G1–G2	4	17.3%	19	82.7%	18	78.3%	5	21.7%	11	78.6%	3	21.4%
G3	7	53.8%	6	46.2%	12	75.5%	4	25.0%	17	77.3%	5	22.7%
				*p* = 0.0065				*p* = 0.81				*p* = 0.25
T classification												
T1–T2	8	29.6%	19	70,4%	21	75.0%	7	25.0%	19	76.0%	6	24.0%
T3–T4	3	33.3%	6	66,7%	9	81.8%	2	18.2%	9	81.8%	2	18.2%
				*p* = 0.96				*p* = 0.65				*p* = 0.57
pN status												
N0	10	43.4%	13	56.6%	20	76.9%	6	23.1%	17	73.9%	6	26.1%
N+	1	7.6%	12	92.4%	10	76.9%	3	23.1%	11	84.6%	2	15.4%
				*p* = 0.04				*p* = 1.00				*p* = 0.46

**Table 3 cancers-15-02461-t003:** Correlation between the three biomarkers (Mann–Whitney test).

PDL1					CD44					AGT7				
	Negative		Positive			Negative		Positive			Negative		Positive	
	27	75.0%	9	25.0%		11	30.6%	25	69.4%		28	77.7%	8	22.3%
AGT7					PDL1					CD44				
Negative	20	71.4%	8	28.6%	Negative	9	33.3%	18	66.7%	Negative	5	50.0%	5	50.0%
Positive	7	87.5%	1	12.5%	Positive	2	22.2%	7	77.8%	Positive	23	88.5%	3	11.5%
*p* = 0.36					*p* = 0.53					*p* = 0.02				

## Data Availability

The data presented in this study are available in this article.

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
