# Peer review of "CD44, PDL1, and ATG7 Expression in Laryngeal Squamous Cell Carcinomas with Tissue Microarray (TMA) Technique: Evaluation of the Potential Prognostic and Predictive Roles"

_cancers, 2023, doi:10.3390/cancers15092461_

Round 1

Reviewer 1 Report

The study is well designed and shows the degree of expression of important proteins by malignant cells. Methodologically, everything was done in accordance with the set goals. A good immunohistochemical method was chosen. The results are adequate, as are the explanations about the significance of the expression of the determined proteins.

Author Response

We thank very much the Reviewer for his/her comment.

Reviewer 2 Report

1. In the “Material and methods” section describe in more detail the IRS score, just like CPS is described so the reader dose not have to look for the reference 10 from the paper…

2. The “Results” section could be broken down into subsections, patients clinical-pathohistological data; CD44 expression;  PDL1 and ATG expression just to have clear view over “Results”

3. In Figure 1 and 2 it is not written the type of staining, I assume CD44 so please insert it in the title of the figure, and also use the same magnification for images like you did in the rest of the figures either 10x or 20x and it would be much better if you can show image of L-IRS in comparison to H-IRS

4. DFS survival curves Figures 3,6, 8 please insert p values

Overall, the paper is well written and easy to follow…

Author Response

1.    In the “Material and methods” section describe in more detail the IRS score, just like CPS is described so the reader dose not have to look for the reference 10 from the paper…
We have specified in the material and methods the IRS evaluation
2.    The “Results” section could be broken down into subsections, patients clinical-pathohistological data; CD44 expression;  PDL1 and ATG expression just to have clear view over “Results”
We have modified as you suggest
3.    In Figure 1 and 2 it is not written the type of staining, I assume CD44 so please insert it in the title of the figure, and also use the same magnification for images like you did in the rest of the figures either 10x or 20x and it would be much better if you can show image of L-IRS in comparison to H-IRS
We have corrected as you indicated
4.    DFS survival curves Figures 3,6, 8 please insert p values
P value has been reported in the figures 3,6,8.

Reviewer 3 Report

Puzzo et al. have presented an interesting comparative study on the role CD44, PDL1 and ATG7 expression on the laryngeal cancer and their usefulness as a biomarker. This patient data research in this article is indeed a valuable information to understand the underlying mechanism of laryngeal squamous cell carcinomas. There are some minor comments for the authors to address.

    1.  Could the authors include the statistical p value of the data in the graph or in its figure captions for Kaplan Meier graphs in figure 3, 6 and 8 so that the reader do not have to refer the texts.

2  2. Could it be concluded that PDL1 could also be a biomarker for this cancer although not as effective as CD44.

     3. As the authors highlight in the article about CD44, it would be good to have a short discussion regarding how CD44 could also be used as a biomarker for cancer targeting as it has been widely explored by researchers (https://doi.org/10.1021/acs.molpharmaceut.2c00439)

Author Response

  1. Could the authors include the statistical p value of the data in the graph or in its figure captions for Kaplan Meier graphs in figure 3, 6 and 8 so that the reader do not have to refer the texts.

P value has been reported in the figure 3,6,8

  1. Could it be concluded that PDL1 could also be a biomarker for this cancer although not as effective as CD44.

We have added your consideration in the conclusion

  1. As the authors highlight in the article about CD44, it would be good to have a short discussion regarding how CD44 could also be used as a biomarker for cancer targeting as it has been widely explored by researchers (https://doi.org/10.1021/acs.molpharmaceut.2c00439)

In the discussion we have included a consideration about the reference suggested and it has been included in the reference list.